# Health-related quality of life and associated factors among patients with stroke at tertiary level hospitals in Ethiopia

**Ashenafi Zemed[1], Kalkidan Nigussie Chala[1], Getachew Azeze Eriku[1], Andualem Yalew Aschalew[2]***

**1** Department of Physiotherapy, School of Medicine, College of Medicine and Health Sciences, University of Gondar, Gondar, Ethiopia, **2** Department of Health Systems and Policy, Institute of Public Health, College of Medicine and Health Sciences, University of Gondar, Gondar, Ethiopia

* yalewandualem@gmail.com

**Data Availability Statement:** All relevant data are within the paper and its Supporting information files.

## Abstract

### Introduction

Evidence on a patient-centered assessment of outcome among patients with stroke is limited in Ethiopia. Therefore, this study aimed to assess the level of health-related quality of life (HRQOL) and associated factors in Ethiopia's tertiary level hospitals.

### Methods

A cross-sectional study was conducted at three tertiary level hospitals (Felege Hiwot comprehensive specialized hospital, University of Gondar comprehensive specialized hospital, and Dessie referral hospital) from April 1 to May 31, 2019. A total of 180 patients with stroke were included, and a consecutive sampling method was employed to recruit the participants. RAND 36-Item Health Survey was used to measure the HRQOL. A generalized linear model with a gamma distribution and log-link function was used to investigate potential predictors, and variables with a $P$ value of <0.05 were considered statistically significant.

### Results

Out of the participants, 50.56% were female. The average age and average duration of illness were 59.04 (12.71) and 1.5 (1.46) years, correspondingly. The physical health domain score was higher than the mental health domain score. Education ($P = 0.041$), social support ($P = 0.050$), disability ($P < 0.001$), co-morbidity ($P = 0.011$), depression ($P = 0.015$) and income (<1000 ETB $P = 0.002$; 1000–4000 ETB $P = 0.009$) were associated with physical health domain. Whereas, ischemic stroke ($P = 0.014$), education ($P = 0.020$), disability ($P < 0.001$), and depression ($P < 0.001$) were associated with the mental health domain.

### Conclusion

The HRQOL of the patients was low. Social support and lower disability status were associated with higher HRQOL, whereas disability and depression were associated with higher HRQOL. Therefore, attention should be given to strengthening social support; health

**Funding:** This study was part of a master thesis funded by the University of Gondar. The preliminary findings of the study were presented at the School of Medicine, University of Gondar.

**Competing interests:** The authors have declared that no competing interests exist.

professionals should focus on reducing disability/physical dependency and depression, as these are vital factors for improving HRQOL.

## Introduction

The global lifetime risk of stroke rose to 25% for those 25 years and over in the last three decades [1]. Notably, stroke remains the second leading cause of death and disability world-wide, with 5.5 million deaths, in 2016. Although stroke incidence decreased in most regions (except East Asia and Southern sub-Saharan Africa), a decline in stroke death rates and the aging population makes it still prevalent [2–4].

Ethiopia has also shared the global problem, and stroke is becoming more prevalent. The Burden of Disease study reported that, in 2016, there were 52,548 incidences of stroke and 38,353 deaths in Ethiopia [5]. Previous studies showed that in-hospital mortality ranged from 11% to 44%. The majority of patients presented with some sort of disability such as weakness of the body, inability to communicate, etc. Moreover, in all studies, the proportion of patients with hypertension, which is a known risk factor for stroke, was high [6–11].

Although stroke causes significant functional sequela, objective assessment approaches such as clinical assessment (neurological function test) and biochemical tests often fail to gauge the subjective experience of the impacts of the disease [12]. However, the patient-report outcome, for instance, health-related quality of life (HRQOL), has increasingly been used as a crucial measurement for assessing the disease's effects from the patient's perspective [13]. HRQOL is a broad-ranging concept incorporating the person's physical health, psychological state, level of independence, social relationships, and personal beliefs in a complex way and focuses on the impact of health status on HRQOL [14]. Thus, it is a subjective appraisal of the patient's current level of functioning and satisfaction with their health compared to what they believe to be ideal. In other words, it is an inherent attribute of self-perception of various aspects of the patient's general health [15,16]. Concepts in stroke care have expanded from decreasing mortality and morbidity to improving the functional level and quality of life [17]. Therefore, the appraisal of patient-reported outcomes is a valuable method, particularly among patients with a chronic disease such as stroke whose psychological and social wellbeing, just as physical health, are affected by the disease.

Several factors have a significant association with HRQOL. The two well-known factors are depression and disability [18–22]. Others, including age [23], social support [24,25], income [19,26], co-morbidities like diabetes and hypertension [22,27,28], and sex [29] were also significantly associated with HRQOL.

Literature about HRQOL has been grown on various diseases and injuries, including stroke. There is, however, a huge gap in Africa, including Ethiopia, in this area. For instance, one study showed that from 50 different studies (1970–2017) on stroke, only ten studies were on HRQOL, and none of them were in Ethiopia [30]. To our knowledge, studies in our country, including in the current study areas, focus on prevalence, risk factor, clinical presentation or profiles of patients, and mortality rate of stroke [11,31–34]. Therefore, this study aimed to determine the level of HRQOL and associated factors (sociodemographic, clinical, and environmental) among patients with stroke at the three tertiary hospitals in the Amhara National Regional State.

## Materials and methods

### Study design and setting

An institutional-based cross-sectional study was conducted at three tertiary level hospitals: Felege Hiwot comprehensive specialized hospital, University of Gondar comprehensive specialized hospital, and Dessie referral hospital, from April 1 to May 31, 2019. The hospitals are found in the Amhara National Regional State, northwest of Ethiopia. Each hospital has a chronic illness follow-up outpatient department (OPD) for chronic diseases, including stroke. The OPD is opened five days a week (Monday up to Friday), and patients with neurologic disorders, such as patients with stroke, get the service two days per week.

### Subjects and sampling technique

A total of 180 patients with stroke were included, and the sample size was allocated proportionally: sixty participants per hospital. The population consisted of patients with stroke who were seeking post-stroke rehabilitation at the chronic illness OPD during the study period. Patients who were involved in the study were selected based on established criteria. Accordingly, after checking the patients' medical record card, all adult patients ($\geq$ 18 years) with any type of stroke who had the disease for three or more months were included, whereas patients with a brain tumor, any musculoskeletal problem, mental disorder, traumatic brain injury, or spinal cord injury were excluded from the study [18,35]. A consecutive sampling method was employed to recruit the participants until the required sample size was reached.

### Data collection tools and procedures

The tool has three sections: sociodemographic variables (sex, age, marital status, educational status, religion, occupation, residence, and income), clinical variables (duration of a stroke, type of stroke, affected site, co-morbidity, and disability), and psychosocial variables (social support and depression). The RAND 36-Item Health Survey was translated into Amharic language and back-translated into the English language to keep its consistency. Furthermore, a pretest was done on five percent of the sample size outside the study area with a similar setting. A semi-structured interviewer-administered questionnaire was used to collect the necessary information. For those who fulfill the inclusion criteria but had communication or cognitive problems, we used reliable proxies to collect the data. Meanwhile, some of the clinical variables (type of stroke, affected side, co-morbidity, and duration of the disease) were collected from the patient's medical record.

**Measurements.** *Health-related quality of life*. The HRQOL was measured by the RAND 36-Item Health Survey (Version 1.0). The tool contains 36 items, which can be computed into eight scales. The scoring has two-steps. First, precoded numeric values were recoded per the scoring manual. Second, items on the same scale were averaged together to create the scores. The higher the scores, the better the HRQOL. Finally, the eight scales were aggregated into two distinct summary measures: a physical component summary (PCS), which represented the physical dimension of HRQOL, and a mental component summary (MCS), which described the mental dimension of HRQOL [36,37]. The validity and reliability of the tools have been demonstrated in Ethiopia [38].

*Depression*. Depression was measured by the Hospital Anxiety and Depression Scale (HADS). There were seven-item self-report questions that measure the level of depression. The scale score ranged from 0 to 21, with 0–7 represent no depression, 8–10 represent borderline depression, and 11–21 represent depression [39].

*Social support*. The Oslo 3-items social support scale was used to measure social support. The tool has three items. A sum index was made by summarising the raw scores and the sum ranging from 3 to 14. A score of 3–8 is "poor social support," 9–11 is "moderate social support," and 12–14 is "strong social support" [40].

*Functional ambulation category*. The functional ambulation category (FAC) is a scale that measures the ambulation ability of the patient. It comprises six categories ranging from 0 represents non-functional to 5 represents normal ambulation [41].

## Data analysis

The collected data were checked for completeness. Then, codes were given to each questionnaire and entered into EpiInfo Version 7 Software. Further analysis was done with Stata version 14. Descriptive statistics were presented using frequencies, percentages, means, and standard deviations. One-way analysis of variance (ANOVA) was used to determine significant differences between mean levels of HRQOL across the three hospitals. Assumptions of ANOVA: homogeneity of variance, normality, and presence of significant outliers were checked. Initially, bivariate analysis was done to identify factors associated with each domain (PCS and MCS) of HRQOL, independently, and variables with *P* value <0.2 [42] were selected for the final model: generalized linear model (GLM) with gamma family and log link function. The ability to handle a larger class of error distributions and data types is a key improvement of GLM over linear models [43,44]. To validate the distribution used in the GLM, the modified park test was applied. We fitted two models consisting of the PCS (model 1) and the MCS (model 2) as a dependent variable. Model one included sex, age, marital status, education status, occupation, income, type of stroke, co-morbidity, depression, disability, duration of a stroke, and social support. Model two included sex, age, education status, occupation, income, type of stroke, depression, disability, social support, and residence. Clinically meaningful combinations of variables and their interactions were assessed for effect; however, not engaged in the last model. The potential for multiple collinearities was tested using the variance inflation factor (VIF); where VIF <10 was desirable. Model adequacy was gauged by a progressive reduction in AIC (Akaike Information Criterion) and BIC (Bayesian Information Criterion) [45]. Finally, variables with a P-values of <0.05 were considered statistically significant.

## Ethical considerations

Ethical clearance was obtained from the Ethical Review Committee of the School of Medicine, College of Medicine and Health Science, University of Gondar (reference number SoM/1238/2019). A permission letter was given to the representatives of the chronic illness OPD. All participants were oriented to the study's objectives and purpose before they participated, and they provided written informed consent. Patients at health facilities were informed that participation had no impact on the provision of their healthcare. Study team members safeguarded the confidentiality and anonymity of study participants throughout the entire study. This study was conducted in accordance with the Declaration of Helsinki.

## Results

### Sociodemographic and clinical characteristics of study participants

A total of 180 patients were interviewed with a 100% response rate. Almost half (50.56%) of them were female, and the mean (SD) age of the participants was 59.04 (12.71) years. Out of the participants, 66.67% were urban dwellers. A significant number of patients (66.77%) were diagnosed as having ischemic stroke. The majority of the participants (74.44%) had co-

morbidity; 117 (87.31%) of them were hypertensive. A considerable amount of patients (80.00%) had some degree of disability. Also, half of the patients had depression (Table 1).

## Health-related quality of life

The RAND 36-Item Health Survey had good internal reliability with Cronbach Alpha of $\alpha$ = 0.93 and $\alpha$ = 0.82 for the physical and the mental domain, respectively. Out of the eight scales, role limitations due to physical health problems (15.69) and emotional problems (19.81) were the lowest. In contrast, the bodily pain scale score was the highest (68.22). Moreover, the PCS score was higher 44.36 (21.15) than the MCS 39.54 (17.13) (Fig 1).

A one-way ANOVA analysis showed no significant differences for the physical and mental components scores of HRQOL among the three hospitals (Table 2). The assumptions of ANOVA were checked and fulfilled: Bartlett's test for equal variances showed that the variance was homogenous. The physical and mental domains of HRQOL were approximately normally distributed (skewness = 0.52 and kurtosis = 2.51) and (skewness = 0.84, and kurtosis = 2.80), respectively. Moreover, after we fitted the regression model, the predicted Cook's distance showed no outliers or influential observations, where Cook's distance less than one was considered appropriate.

## Factors associated with HRQOL

We fitted two models consisting of the PCS (model 1) and the MCS (model 2) as outcome variables. GLM was fitted to identify factors associated with the HRQOL. The modified Park test result showed that the HRQOL score was within the gamma distribution.

**Factors associated with the physical component summary.** It was found that individuals who were grade 9–12 had 22% higher HRQOL than individuals who were unable to read and write (exp(b) = 1.22, P = 0.041). HRQOL increased by about 3% with each additional score of social support (exp(b) = 1.03, P = 0.05). In addition, as the disability status improved, the physical component of HRQOL increased by about 20% (exp(b) = 1.20, P<0.001). Whereas, individuals with co-morbidity had 15% lower HRQOL than individuals who had co-morbidity (exp(b) = 0.85, P = 0.011); individuals who had income less than ETB 1,000 and ETB 1,000–4,000 had 21% and 18% lower HRQOL (exp(b) = 0.79, P = 0.002; exp(b) = 0.82, P = 0.009, respectively) than individuals who had income greater than ETB 4,000. Besides, HRQOL was reduced by about 2% with each additional score of depression (exp(b) = 0.98, P = 0.015) (Table 3).

**Factors associated with the mental component summary.** In the second model, variables that had an association with the MCS were education, disability, stroke type, and depression.

It was found that individuals who were grade 9–12 had 22% higher HRQOL than individuals who were unable to read and write (exp(b) = 1.22, P = 0.020). Moreover, as the disability status improved, the mental component of HRQOL increased by about 12% (exp(b) = 1.12, P<0.001). However, HRQOL was reduced by 3% with each additional score of depression (exp(b) = 0.97, P<0.001), and patients with ischemic stroke had 12% lower HRQOL than individuals who had a hemorrhagic stroke (exp(b) = 0.88, P = 0.014) (Table 4).

## Discussion

This study aimed to assess the HRQOL and associated factors among patients with stroke at the three tertiary level hospitals. The RAND 36-Item Health Survey, which has eight scales, was used to measure the HRQOL. The scale's scores ranged from 15.69 (physical health) to

**Table 1. Sociodemographic and clinical characteristics of study participants (N = 180).**

| Variables | Frequency (%) | Mean (SD) |
|---|---|---|
| **Sex** | | |
| Male | 89 (49.44) | |
| Female | 91 (50.56) | |
| **Age in years** | | 59.04 (12.71) |
| **Residence** | | |
| Urban | 120 (66.67) | |
| Rural | 60 (33.33) | |
| **Marital status** | | |
| Single | 11 (6.11) | |
| Widowed | 27 (15.00) | |
| Divorced | 42 (23.33) | |
| Married | 100 (55.56) | |
| **Religion** | | |
| Orthodox | 98 (54.44) | |
| Muslim | 66 (36.67) | |
| Protestant | 16 (8.89) | |
| **Occupation** | | |
| Government employed | 38 (21.11) | |
| Private employed | 58 (32.22) | |
| Housewife | 50 (27.78) | |
| Farmer | 27 (15.00) | |
| Retired | 7 (3.89) | |
| **Educational status** | | |
| Unable to read and write | 63 (35.00) | |
| 1–8 Grade | 47 (26.11) | |
| 9–12 Grade | 29 (16.11) | |
| College and above | 41 (22.78) | |
| **Income** | | |
| ETB <1,000 | 76 (42.22) | |
| ETB 1,000–4,000 | 41 (22.78) | |
| ETB >4,000 | 63 (35.00) | |
| **Type of stroke** | | |
| Ischemic | 120 (66.77) | |
| Hemorrhagic | 60 (33.33) | |
| **Duration of stroke in years** | | 1.55 (1.46) |
| **Co-morbidity** | | |
| Yes | 134 (74.44) | |
| No | 46 (25.56) | |
| **Type of co-morbidity** | | |
| HTN | 117 (87.31) | |
| DM | 4 (2.99) | |
| DM & HTN | 8 (5.97) | |
| Cardiac & HTN | 5 (3.73) | |
| **Affected side** | | |
| Right | 96 (53.33) | |
| Left | 84 (46.67) | |
| **Social support** | | 10.47 (2.38) |
| Poor | 40 (22.22) | |

(*Continued*)

**Table 1.** (Continued)

| Variables | Frequency (%) | Mean (SD) |
|---|---|---|
| **Intermediate** | 75 (41.67) | |
| **Strong** | 65 (36.11) | |
| **Depression** | | 7.85 (4.51) |
| **Normal** | 91 (50.56) | |
| **Borderline** | 46 (25.56) | |
| **Depressed** | 43 (23.88) | |
| **FAC** | | 2.91 (1.63) |
| **Non-functional** | 21 (11.67) | |
| **Nonfunctional ambulation** | 17 (9.44) | |
| **Household ambulation** | 33 (18.33) | |
| **Surroundings of the house ambulation** | 32 (17.78) | |
| **Independent community ambulation** | 41 (22.78) | |
| **Normal ambulation** | 36 (20.00) | |

ETB, Ethiopia birr; FAC, functional ambulation category; HTN, hypertension; DM, diabetes mellitus; SD, standard deviation.

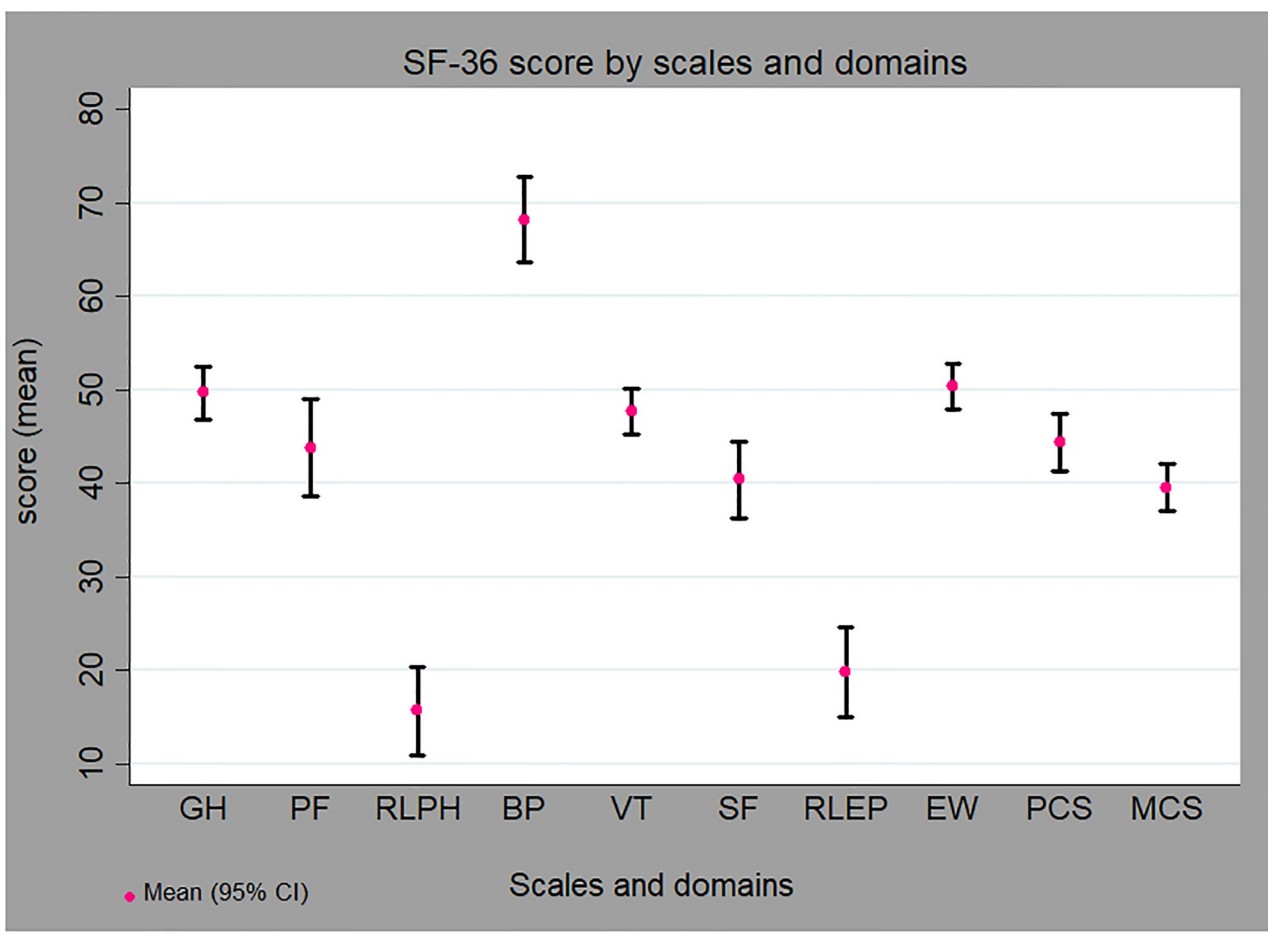

**Fig 1. HRQOL scales and domains score of patients with stroke.** SF-36 scales: GH, General health perceptions; PF, Physical functioning; RLPH, Role limitations due to physical health problems; BP, Bodily pain; VT, Fatigue; SF, Social functioning; RLEP, Role limitations due to emotional problems; EW, Emotional wellbeing. SF-36 domains: PCS, Physical component score; MCS, Mental component score.

**Table 2. Comparisons of HRQOL scores among the three hospitals (N = 180).**

| HRQOL domains | UoGCSH, Mean (SD) | FHCSH, Mean (SD) | DRH, Mean (SD) | P-value [a] |
|---|---|---|---|---|
| PCS | 41.01 (22.30) | 45.42 (21.01) | 46.64 (20.00) | 0.309 |
| MCS | 41.39 (17.03) | 39.86 (18.30) | 37.37 (16.02) | 0.433 |

UoGCSH, University of Gondar comprehensive and specialized hospital; FHCSH, Felege Hiwot comprehensive and specialized hospital; DCSH, Dessie referral hospital.
[a]P < 0.05.
PCS: Bartlett's test for equal variances: chi2(2) = 0.6981 Prob >chi2 = 0.705.
MCS: Bartlett's test for equal variances: chi2(2) = 1.0365 Prob >chi2 = 0.596.

68.22 (bodily pain). The MCS score was lower compared to the PCS score. Moreover, education, depression, and disability were significantly associated with both domains.

Stroke affects the patient's physical, mental, and social aspects of life [46,47]. Measurement of patient-report outcome is increasingly important [48]. This study tried to measure the impact of stroke on the patient's HRQOL. The findings revealed that most of the eight scales' scores were low. The scales most affected were role limitations due to physical health and emotional health problems. Evidence showed that a sudden interruption of blood resulted in a stroke that damages brain tissues, which leads to a cognitive problem, disability, and emotional disturbance such as depression [27,46,49]. This finding also supported that more than 57% of patients were physically non-functional or had some degree of limited mobility, and around 50% of them had depression. These might lead to cutting down the time the patients spent on work or other activities and accomplished less than they would like, which reduces their roles and HRQOL.

Moreover, social functioning and physical functioning were affected next to role limitation due to a stroke. As we mentioned above, disability and depression are common among patients with stroke, including the patient's day-to-day activities, social engagement, and interaction. A previous study showed that patients with a physical disability had limited daily activity and poor community reintegration [49].

Our study found better patient HRQOL as compared to the studies in Egypt [21,50]. Plausible explanations might be a difference in sample size and characteristics of the participants. For instance, in those studies, most participants had co-morbidity and dependent on daily activities. However, the current study is lower than the studies in Bangkok, Thailand [35], Brazil [15], New Zealand [51], in all scales. In addition, a study in Ohio [27] has a higher HRQOL score than the current study for most of the scales. A plausible explanation could be that those studies included cognitively fitted patients; most were from a rehabilitation center. They had a long duration of the event that helps the patients recover well psychologically and physically.

The second objective of this study was to identify factors that had an association with the HRQOL. Patients with good social support had a higher score on PCS than those who had poor social support. Previous studies are in line with this finding [20,25,52]. Stroke has an impact on the physical health of the patients. For instance, disability after a stroke makes the patient life dependent on others (families) [53,54]. This indicated that social support is important in stroke rehabilitation, especially after discharge to home. Thus, advice and encouragement from families, friends, or religious fathers might improve the patient's perception of wellbeing and functioning; consequently, enhance the patient's HRQOL.

This study showed that non-functional or limited mobility patients had lower HRQOL on both domains (PCS and MCS). This result is consistent with those of previous studies [18,21,55,56]. As the status of the patient's mobility improves, their HRQOL also enhanced. According to a study from Prishtina, Kosovo, as the physical state improves, it becomes easier

**Table 3. Generalized linear model analysis to identify factors associated with PCS among patients with stroke (N = 180).**

| Variables | exp(b) | 95% CI | *P* value |
|---|---|---|---|
| **Gender** | | | |
| Female | 1.02 | 0.89, 1.16 | 0.813 |
| Male | ref | ref | ref |
| **Age** | 1.00 | 0.99, 1.00 | 0.445 |
| **Marital status** | | | |
| Single | 0.87 | 0.68, 1.11 | 0.278 |
| Widowed | 0.88 | 0.75, 1.03 | 0.106 |
| Divorced | 0.89 | 0.78, 1.02 | 0.087 |
| Married | ref | ref | ref |
| **Occupation** | | | |
| Private employed | 1.10 | 0.92, 1.31 | 0.290 |
| Housewife | 1.18 | 0.95, 1.46 | 0.139 |
| Farmer | 1.12 | 0.91, 1.38 | 0.299 |
| Retired | 1.23 | 0.90, 1.68 | 0.192 |
| Government employed | ref | ref | ref |
| **Education status** | | | |
| 1–8 Grade | 1.13 | 0.98, 1.31 | 0.091 |
| 9–12 Grade | 1.22 | 1.01, 1.47 | 0.041 |
| College & above | 1.05 | 0.85, 1.30 | 0.666 |
| Unable to read and write | ref | ref | ref |
| **Income** | | | |
| ETB <1,000 | 0.79 | 0.69, 0.92 | 0.002 |
| ETB 1,000–4,000 | 0.82 | 0.71, 0.95 | 0.009 |
| ETB >4,000 | ref | ref | ref |
| **Co-morbidity** | | | |
| Yes | 0.85 | 0.74, 0.96 | 0.011 |
| No | ref | ref | ref |
| **Type of stroke** | | | |
| Ischemic | 0.97 | 0.86, 1.10 | 0.653 |
| Hemorrhagic | ref | ref | ref |
| **Social support** | 1.03 | 1.00, 1.05 | 0.050 |
| **Depression** | 0.98 | 0.97, 1.00 | 0.015 |
| **FAC** | 1.20 | 1.15, 1.25 | 0.000 |
| **Duration** | 1.02 | 0.98, 1.06 | 0.270 |
| **Constant** | 24.93 | 15.83, 39.27 | 0.000 |

CI, confidence interval; ETB, Ethiopia birr; FAC, functional ambulation category; SD, standard deviation.

to reintegrate into the community [49]. Another significant predictor of HRQOL was depression, one of the emotional problems that affect function level. Several studies have shown that depression influences the patient's HRQOL [18,19,28,55,57]. The current study is also in line with the above studies that depression reduces the mental domain of HRQOL. A study by Chang et al showed emotional status is a significant predictor of functional level and HRQOL at the chronic stage among stroke survivors [58].

Likewise, education was associated positively with both domains. Patients who achieved up to high schools had a better HRQOL than those who could not read and write. Studies in

**Table 4. Generalized linear model analysis to identify factors associated with MCS among patients with stroke (N = 180).**

| Variables | exp(b) | 95% CI | *P* value |
|---|---|---|---|
| **Gender** | | | |
| Female | 1.01 | 0.90, 1.13 | 0.833 |
| Male | ref | ref | ref |
| **Age** | 1.00 | 1.00, 1.01 | 0.268 |
| **Occupation** | | | |
| Private employed | 0.99 | 0.86, 1.15 | 0.931 |
| Housewife | 1.09 | 0.90, 1.31 | 0.376 |
| Farmer | 1.14 | 0.94, 1.37 | 0.174 |
| Retired | 1.13 | 0.86, 1.47 | 0.384 |
| Government employed | ref | ref | ref |
| **Education status** | | | |
| 1–8 Grade | 1.08 | 0.96, 1.23 | 0.206 |
| 9–12 Grade | 1.22 | 1.03, 1.43 | 0.020 |
| College & above | 1.14 | 0.95, 1.37 | 0.169 |
| Unable to read and write | ref | ref | ref |
| **Residence** | | | |
| Rural | 1.00 | 0.90, 1.12 | 0.993 |
| Urban | ref | ref | ref |
| **Income** | | | |
| ETB <1,000 | 0.97 | 0.86, 1.10 | 0.620 |
| ETB 1,000–4,000 | 0.94 | 0.83, 1.07 | 0.366 |
| ETB >4,000 | ref | ref | ref |
| **Type of stroke** | | | |
| Ischemic | 0.88 | 0.79, 0.97 | 0.014 |
| Hemorrhagic | ref | ref | ref |
| **Social support** | 0.99 | 0.97, 1.02 | 0.591 |
| **Depression** | 0.97 | 0.96, 0.98 | 0.000 |
| **FAC** | 1.12 | 1.08, 1.16 | 0.000 |
| **Duration** | 1.00 | 0.97, 1.03 | 0.991 |
| **Constant** | 32.86 | 22.16, 48.72 | 0.000 |

CI, confidence interval; ETB, Ethiopia birr; FAC, functional ambulation category.

There was no significant association between each domain of quality of life with sex, affected side, marital status, occupation, duration of stroke, and residence.

Kenya and Ohio [27,29] are in line with this finding. Education is an essential factor in understanding self-care management. Patients with a high educational level can easily read and understand stroke's effects, leading to better awareness about the disease. Furthermore, it contributes to a high rate of adherence to self-care management.

The presence of co-morbidity appears to contribute toward lower PCS compared with those who had not co-morbidity. Other studies are also in line with this study [27,28]. The commonest co-morbidities among stroke patients are HTN and DM. In this study, 74.44% of patients had co-morbidities, and HTN was the most prevalent. The finding is compatible with studies conducted in Ethiopia [6,11,31,59].

Participants who had a lower income had lower PCS scores than those who had a higher income (ETB >4,000). Previous studies have also reported that low income or poor

socioeconomic status reduced the patient's HRQOL [19,26]. Another study also showed disability, especially severe form, needs more cost for treatment [13]. However, patients with low income could not afford sufficient treatment and rehabilitation. Although early initiation of rehabilitation is essential for functional recovery, low-income causes delayed rehabilitation initiation or not at all result in poor physical health [60].

There is controversy regarding the relationship between stroke type and HRQOL. Our study demonstrated that patients with ischemic stroke were associated with a lower quality of life concerning mental health than hemorrhagic stroke patients. The study in Egypt is also consistent with this finding [50]. However, other studies have shown that type of stroke did not associate with HRQOL [15,35,61].

There was no significant association between each domain of quality of life and sex, affected side, marital status, and occupation. Literature had inconsistent findings due to different methods and tools for measurement, subject selection criteria, and sociocultural differences.

## Limitation of the study

The limitation of this study is that it is difficult to show causal relationships due to the study's cross-sectional nature. In addition, relatively a small sample size who represent only stroke survivors sought care at public hospitals. Therefore, a further longitudinal study with a larger sample size should be done to investigate the influential factors on stroke survivors' HRQOL. However, the strength is that we used a multicenter and multi-dimensional tool for the assessments of HRQOL.

## Conclusion

This study indicated that a stroke had a remarkable impact on patients' HRQOL. Notably, the mental dimension of HRQOL was affected more. The HRQOL of patients with depression was found to be more affected. Additionally, having co-morbidities and low income had a negative effect on the physical domain, while higher education status, lower disability status, and social support showed a positive effect. Therefore, attention should be given to strengthening social support; health professionals should focus on reducing disability/physical dependency and depression, as these are vital factors for improving HRQOL. Moreover, this study should be used as a starting point for further studies.

## Supporting information

**S1 Dataset. HRQOL (dataset).**
(DTA)

## Acknowledgments

We are very thankful to the University of Gondar for the approval of the ethical issue and its technical support. We forward our appreciation to the hospital managers for allowing us to conduct this research and their cooperation. Finally, we would like to thank the study participants for their volunteer participation and also data collectors and supervisors for their genuineness and quality of work during data collection.

## Author Contributions

**Conceptualization:** Ashenafi Zemed, Kalkidan Nigussie Chala, Getachew Azeze Eriku, Andualem Yalew Aschalew.

**Data curation:** Ashenafi Zemed.

**Formal analysis:** Ashenafi Zemed, Getachew Azeze Eriku, Andualem Yalew Aschalew.

**Funding acquisition:** Ashenafi Zemed.

**Investigation:** Ashenafi Zemed.

**Methodology:** Ashenafi Zemed, Andualem Yalew Aschalew.

**Project administration:** Ashenafi Zemed.

**Resources:** Ashenafi Zemed.

**Software:** Ashenafi Zemed, Andualem Yalew Aschalew.

**Supervision:** Ashenafi Zemed, Kalkidan Nigussie Chala, Getachew Azeze Eriku, Andualem Yalew Aschalew.

**Validation:** Ashenafi Zemed.

**Visualization:** Ashenafi Zemed, Andualem Yalew Aschalew.

**Writing – original draft:** Ashenafi Zemed, Andualem Yalew Aschalew.

**Writing – review & editing:** Ashenafi Zemed, Kalkidan Nigussie Chala, Getachew Azeze Eriku, Andualem Yalew Aschalew.

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
