## [Decision Letter · Decision Letter 0]

13 Aug 2020

PONE-D-20-19104

Health-related quality of life and associated factors among patients with stroke at the tertiary level hospitals in Ethiopia

PLOS ONE

Dear Dr. Aschalew,

Thank you for submitting your manuscript to PLOS ONE. After careful consideration, we feel that it has merit but does not fully meet PLOS ONE’s publication criteria as it currently stands. Therefore, we invite you to submit a revised version of the manuscript that addresses the points raised during the review process.

We look forward to receiving your revised manuscript.

Kind regards,

Amir H. Pakpour, Ph.D.

Academic Editor

PLOS ONE

Journal Requirements:

2. Your ethics statement must appear in the Methods section of your manuscript. If your ethics statement is written in any section besides the Methods, please move it to the Methods section and delete it from any other section. Please also ensure that your ethics statement is included in your manuscript, as the ethics section of your online submission will not be published alongside your manuscript.

3.Thank you for stating the following in the Funding Section of your manuscript:

[This study was part of a master thesis funded by the University of Gondar. The preliminary findings of the study were presented at the School of Medicine, University of Gondar.]

 [The funders had no role in study design, data collection and analysis, decision to publish, or preparation of the manuscript.]

Reviewers' comments:

Reviewer's Responses to Questions

**Comments to the Author**

1. Is the manuscript technically sound, and do the data support the conclusions?

Reviewer #1: Yes

2. Has the statistical analysis been performed appropriately and rigorously? 

Reviewer #1: No

3. Have the authors made all data underlying the findings in their manuscript fully available?

Reviewer #1: No

4. Is the manuscript presented in an intelligible fashion and written in standard English?

Reviewer #1: No

5. Review Comments to the Author

Reviewer #1: Dear author(s),

Many thanks for choosing this subject. I appreciate you because of all efforts, short of some recommendation below:

1. In your abstract, please do not change the subtitles. I mean Purpose. Write introduction instead of Purpose.

2. In abstract, please mention to the statistical population and the sample and the method of sampling. It is vital that author(s) present the area I which the study was carried out.

3. Your results in you abstract needs substantial changes. Please follow the routine methods to write this part. It is incomplete and the SD and p-value should be given where the estimations are reported.

4. There are a few keywords please extract further related keywords from https://www.ncbi.nlm.nih.gov/mesh.

5. About your introduction, please elaborate a little bit on HRQOL in the other areas and the importance of this subject. Some parts of Introduction part are related to Discussion part, so please transform them into Discussion part.

6. Why was proportional sampling chosen? And please state the type of sampling.

7. Is there any eligibility criteria? Please cite to related references where the inclusion and exclusion are confirmed.

8. Why was semi-structured interview opted? These types of interviews likely increase the personal bias. Please explain whether this conduction is solo or not.

9. Why was consecutive sampling done while your sampling is proportional? Bias can also occur in consecutive sampling when consecutive samples have some common similarity and it is a non-probability sampling.

10. Your instrument part is too long. To avoid prolixity, please explicate the main items briefly and cite to most related manuscript which can confirm what have been done.

11. Were the presumptions of ANOVA and multiple regression models checked?

12. Was there any outlier or influential observation in your data?

13. Was there any multicollinearity among main variable?

14. Why is a nominal p-value considered before modelling? Is there any valid source to cite? What is the type of model selection? Which kinds of model adequacy checking criteria were chosen?

15. What is the ground to separate two models for physical and mental component? Is there any previous knowledge to do that?

16. Please follow the general writing rules to write the figures. Write the figures with equal number of decimal.

17. Generalized linear model (GLM) is broadly used to analyze data in which there are some categorical predictors or covariates. Was this approach applied? This approach is different from multiple linear regression models, so please state the correct method.

18. Please interpret the effect of predictors clearly and completely and is there any moderator effect upon the related response variable?

19. Why was cut-off considered for some of covariates, for instance, income and education? You definitely lost a part of information by categorizing the continuous variables.

20. You have checked the relations between predictors and response variables and response comprises of 3 components. How did you examine the effects of each component? Which component is the origin of effect in each model? How you convince these parts of effects?

21. There is not any plot or path diagram which can illustrate the path between variables. Moreover, there is not any plot to for descriptive analysis. It is a plus point for your manuscript to display the effects schematically.

With regard to these comments, I thoroughly recommend author(s) revise their paper majorly.

Regards,

Maryam Ganji

6. PLOS authors have the option to publish the peer review history of their article (what does this mean?). If published, this will include your full peer review and any attached files.

Reviewer #1: No

---

## [Author Response · Author response to Decision Letter 0]

29 Sep 2020

Date: September 26, 2020 

Subject: Response to reviewers 

Manuscript title: Health-related quality of life and associated factors among patients with stroke at tertiary level hospitals in Ethiopia

Manuscript ID: PONE-D-20-19104

Dear Dr. AmirH. Pakpour , 

We appreciate the efforts by you and the reviewers on the manuscript. We thank you and the reviewers for the thorough reading and constructive comments and questions of our manuscript and for the opportunity to revise and resubmit. We are pleased to submit the improved research article, including re-analysis, language editing and formatting according to the journal requirement, “Health-related quality of life and associated factors among patients with stroke at tertiary level hospitals in Ethiopia” for your consideration in PLOS ONE. On the following pages, you will find our response to reviewer comments. On behalf of my co-authors, I thank you for your consideration of this resubmission. We appreciate your time and look forward to your response.

Journal Requirements:

1. Please ensure that your manuscript meets PLOS ONE's style requirements, including those for file naming. The PLOS ONE style templates can be found at https://journals.plos.org/plosone/s/file?id=wjVg/PLOSOne_formatting_sample_main_body .pdf and 

Response: 

We have reformatted the manuscript according to the above style guidelines, including file naming.

2. Your ethics statement must appear in the Methods section of your manuscript. If your ethics statement is written in any section besides the Methods, please move it to the Methods section and delete it from any other section. Please also ensure that your ethics statement is included in your manuscript, as the ethics section of your online submission will not be published alongside your manuscript.

Response: 

The ethics statement of the manuscript is within the methods section.

3.Thank you for stating the following in the Funding Section of your manuscript: 

[This study was part of a master thesis funded by the University of Gondar. The preliminary findings of the study were presented at the School of Medicine, University of Gondar.]

We note that you have provided funding information that is not currently declared in your Funding Statement. However, Funding information should not appear in the Acknowledgments section or other areas of your manuscript. We will only publish funding information present in the Funding Statement section of the online submission form.

[The funder had no role in study design, data collection and analysis, decision to publish, or preparation of the manuscript.]

Response:

We have removed the Funding Statement in the manuscript, and we have agreed with the above Funding Statement.

 

Response to Reviewers

Reviewer #1: Dear author(s),

Many thanks for choosing this subject. I appreciate you because of all efforts, short of some recommendation below:

Thank you very much for your comprehensive comments and constructive suggestions. We read and consider each comment very carefully, and thoroughly revise the manuscript according to your comments and suggestions. We hope that the manuscript reads more convincingly after the revision.

1. In your abstract, please do not change the subtitles. I mean Purpose. Write introduction instead of Purpose.

Response: 

Dear reviewer, thank you for your comment, we have changed the subtitle “purpose” into “introduction”. (page 2, line number 13). 

2. In abstract, please mention to the statistical population and the sample and the method of sampling. It is vital that author(s) present the area I which the study was carried out.

Response: 

Thank you for your comment, we have included the study area, sample and sampling methods (page 2, line number 16-19).

3. Your results in you abstract needs substantial changes. Please follow the routine methods to write this part. It is incomplete and the SD and p-value should be given where the estimations are reported.

Response: 

Thanks for pointing out our shortcomings! In this revision, we have included some descriptive, coefficient of significant variables and p-value in the result section of the abstract. 

4. There are a few keywords please extract further related keywords from https://www.ncbi.nlm.nih.gov/mesh.

Response: 

Dear reviewer, thank you for your constructive comment; we have added some keywords. Please, see the revised manuscript line 37.

5. About your introduction, please elaborate a little bit on HRQOL in the other areas and the importance of this subject. Some parts of Introduction part are related to Discussion part, so please transform them into Discussion part.

Response: 

Dear, thank you for your constructive comments. We have given some elaboration on HRQOL (line 54-57), and the importance of studying these subjects (line 61-61 and 68-73). We have also rewritten paragraphs three and four of the introduction section, and some statements were transformed into the discussion section. 

6. Why was proportional sampling chosen? And please state the type of sampling.

Response:

Thank you for the question. One major advantage of proportional sampling is that it produces a sample size that is representative of the size of the groups within the population. Therefore, we use proportional allocation to increase the representativeness of each hospital. We proportionally allocate the sample size to the hospitals, and we used consecutive sampling to draw the study units.

7. Is there any eligibility criteria? Please cite to related references where the inclusion and exclusion are confirmed.

Response: 

Thank you! We have already mentioned the exclusion and inclusion criteria. But based on your suggestion, we have cited two references in the revised manuscript. Please see reference numbers 18 and 35, line 91.

8. Why was semi-structured interview opted? These types of interviews likely increase the personal bias. Please explain whether this conduction is solo or not.

Response:

Thank you for your important question. We used semi-structured interview because some of the variables like age, duration of stroke and type of co-morbidity need such type of formats. But the majority of the questions were structured and semi-structured were not the only type of interview. 

9. Why was consecutive sampling done while your sampling is proportional? Bias can also occur in consecutive sampling when consecutive samples have some common similarity and it is a non-probability sampling.

Response:

Dear reviewer, thank you for your question. Consecutive sampling was used to draw the study units while proportional allocation was applied to increase the representativeness of each hospital. Related to the bias of the sampling, even if the consecutive sampling seems non-probability sampling, we have two justifications for the robustness of the sampling process. 

1. The time of data collection was selected randomly from twelve months.

2. For each visit, patients were visited (appointed) randomly. That means, during the data collection period, we found a mixture of patients, which visited the facility randomly. 

10. Your instrument part is too long. To avoid prolixity, please explicate the main items briefly and cite to most related manuscript which can confirm what have been done.

Response:

Thank you for your comments. In response to your concerns about the length of instrument part we have substantially shortened it by 9 lines. Please, see line numbers 106-124 on page 7. 

11. Were the presumptions of ANOVA and multiple regression models checked?

Response:

Thank you for your question. The outcome variable was to some extent skewed to the right (with skewness of 0.52 for PCS and 0.84 for MCS). One-way ANOVA only requiring approximately normal data because it is quite "robust" to violations of normality, meaning that assumption can be a little violated and still provide valid results. But other important assumptions: homogeneity of variance, comparable observation among the groups and absence of outlier were fulfilled, and these has been described in the result section of HRQOL. Homogeneity of variance was checked by Bartlett's test for equal variances and this can be found below table 2. Please, see line 181-185 and table 2.

Related to the assumption of multiple linear regression models (MLR), in the previously submitted manuscript, we have checked the assumptions of MLR: linearity, normality, multicollinearity, and homoscedasticity. However, the homoscedasticity assumption was not satisfied with the Breusch–Pagan/Cook–Weisberg test Prob > chi2 = 0.0281 and Prob > chi2 = 0.0004 for PCS (model one) and MCS (model two), respectively. Even if we did not explain, robust regression was used as a treatment for heteroscedasticity. But after revision, we perceived robust regression should not be the first choice in such scenario, and transformation should come first. For our data, we found logarithmic transformation is appropriate. However, this transformation has some drawbacks such as retransformation bias. Based on your suggestion and our understanding, we opted generalized linear model (GLM) since it can correct for heteroscedastic errors and do not need to be adjusted for retransformation bias. Therefore, in the revised manuscript GLM with gamma family and log link function was used, and multicollinearity, family distribution and model adequacy were checked and fulfilled. All the changes have been described in the data analysis (lines 135-139) and factors associated (lines 193-196) sections of the revised manuscript. 

12. Was there any outlier or influential observation in your data?

Response: 

Dear, thank you for your important question. We have predicted Cook’s distance after we fitted the regression model and the result showed that there were no outliers or influential observations, where Cook’s distance less than one was considered as appropriate. 

13. Was there any multicollinearity among main variable?

Response:

Thank you for your question. No, the overall variance inflation factor were 1.27 and 1.30 for model one and model two, respectively. Therefore, there was no multicollinearity among the covariates.

14. Why is a nominal p-value considered before modelling? Is there any valid source to cite? What is the type of model selection? Which kinds of model adequacy checking criteria were chosen?

Response: 

Dear, thank you for your important questions. We used nominal p-value to be liberal in the selection of variables. In the other words, we are interested to include marginal confounders and important variables in the final model. Related to the source, we got the information from a book called “Regression Methods in Biostatistics. Authors Eric Vittinghoff, David V. Glidden, Stephen C. Shiboski, Charles E. McCulloch. Chapter 10, page 409” and Maldonado, G., & Greenland, S. (1993). Simulation study of confounder-selection strategies. Am J Epidemiol, 138(11), 923-936, doi:10.1093/oxfordjournals.aje.a116813..

 Moreover, there are plenty of articles that used such a method of variable selection. Here below are some of them, and it's the commonly used cutoff in our setup.

Abebe SM, Berhane Y, Worku A. Barriers to diabetes medication adherence in North West Ethiopia. Springerplus. 2014 Dec 1;3(1):195.

Worku A, Abebe SM, Wassie MM. Dietary practice and associated factors among type 2 diabetic patients: a cross sectional hospital based study, Addis Ababa, Ethiopia. SpringerPlus. 2015 Dec 1;4(1):15.

Amberbir A, Woldemichael K, Getachew S, Girma B, Deribe K. Predictors of adherence to antiretroviral therapy among HIV-infected persons: a prospective study in Southwest Ethiopia. BMC public health. 2008 Dec 1;8(1):265.

The method of model selection is called Selecting Predictors on Statistical Grounds, specifically bivariate screening (analysis): candidate predictors are evaluated one at a time in single predictor models. In some cases, all predictors that meet the screening criterion are included in the final model. For the last question, we have used AIC (Akaike Information Criterion) and BIC (Bayesian Information Criterion) model adequacy criteria.

15. What is the ground to separate two models for physical and mental component? Is there any previous knowledge to do that?

Response:

Thank you for the questions. Yes, there are so many articles that handle SF-36 (HRQOL) as physical and mental. Here below are some of them

Mahran SA, Abdulrahman MA, Janbi FS, Jamalellail RA. The health-related quality of life in stroke survivors: clinical, functional, and psychosocial correlate. Egyptian Rheumatology and Rehabilitation. 2015 Oct 1;42(4):188.

Serda EM, Bozkurt M, Karakoc M, Çağlayan M, Akdeniz D, Oktayoğlu P, Varol S, Kemal NA. Determining quality of life and associated factors in patients with stroke. Turk Soc Phys Med Rehabil. 2015;61:148-54.

 In this study, our dependent variable, HRQOL, has two domains: physical health and mental health. As we mentioned in the instrument section the SF-36 tool can be computed into two summary scales. Therefore, we can fit two independent models for each domain.

16. Please follow the general writing rules to write the figures. Write the figures with equal number of decimal.

Response: 

Agreed. We have tried to follow the general rule, for instance, use numerals for large numbers (say, those over 10) but words for small numbers. Moreover, the figure with decimals was written consistently. 

17. Generalized linear model (GLM) is broadly used to analyze data in which there are some categorical predictors or covariates. Was this approach applied? This approach is different from multiple linear regression models, so please state the correct method. 

Response: 

Thank you for pointing out this appropriate model. We have used GLM in the revised manuscript, and the detailed answer can be found under the response for question 11. 

18. Please interpret the effect of predictors clearly and completely and is there any moderator effect upon the related response variable?

Response:

Thank you for your comments. In the revised manuscript, we have interpreted the effect of each predictor completely as well as we have tested the presence of interaction among predictors and unfortunately none of them were significant. 

19. Why was cut-off considered for some of covariates, for instance, income and education? You definitely lost a part of information by categorizing the continuous variables.

Response:

Thank you for your question. We have revised some covariates (social support, depression and functional ambulation classification) in the model as a continuous variable. However, the variable income and education were collected as stated in the manuscript. In the other words, we are unable to change into continuous variables. Therefore, for your question “why was cut-off considered for some of the covariates?” based on our literature review those variable was considered as categorical and we used them as categorical in order to compare with other works. 

20. You have checked the relations between predictors and response variables and response comprises of 3 components. How did you examine the effects of each component? Which component is the origin of effect in each model? How you convince these parts of effects?

Response:

Dear reviewer, thank you for your comments. If we are not mistaken, you want to say two components: PCS and MCS. These were the outcome variable and we predicted the effect of sociodemographic and clinical variables on the physical and mental dimension of HRQOL. The physical domain was the dependent variable for model one and the mental domain was the dependent variable. The detail can be found in the data analysis section of the manuscript.

21. There is not any plot or path diagram which can illustrate the path between variables. Moreover, there is not any plot to for descriptive analysis. It is a plus point for your manuscript to display the effects schematically. With regard to these comments, I thoroughly recommend author(s) revise their paper majorly

Response:

Dear, thank you. You have raised a very important comment. We have prepared a plot (fig 1) for the descriptive statistics of HRQOL. Related to path diagram (path analysis), it is a good comment, but our objective was to determine the level of HRQOL and associated factors. In order to do path analysis, it needs an independent literature review of mediator variables and another method of analysis such as the structural equation model. With great acknowledgment, we might consider this for another time.

---

## [Decision Letter · Decision Letter 1]

16 Dec 2020

PONE-D-20-19104R1

Health-related quality of life and associated factors among patients with stroke at tertiary level hospitals in Ethiopia

PLOS ONE

Dear Dr. Andualem Aschalew, 

Thank you for submitting your manuscript to PLOS ONE. After careful consideration, we feel that it has merit but does not fully meet PLOS ONE’s publication criteria as it currently stands. Therefore, we invite you to submit a revised version of the manuscript that addresses the points raised during the review process.

You have addressed most of the reviewer's comments. However, the manuscript needs further revision. Please revise the data analysis section and abstract as suggested by the reviewer.

We look forward to receiving your revised manuscript.

Kind regards,

Anandh Babu Pon Velayutham

Academic Editor

PLOS ONE

Additional Editor Comments (if provided):

The authors addressed most of the reviewer's comments. However, the manuscript needs further revision. Please revise the data analysis section and abstract as suggested by the reviewer.

Reviewers' comments:

Reviewer's Responses to Questions

**Comments to the Author**

1. If the authors have adequately addressed your comments raised in a previous round of review and you feel that this manuscript is now acceptable for publication, you may indicate that here to bypass the “Comments to the Author” section, enter your conflict of interest statement in the “Confidential to Editor” section, and submit your "Accept" recommendation.

Reviewer #1: (No Response)

2. Is the manuscript technically sound, and do the data support the conclusions?

Reviewer #1: Yes

3. Has the statistical analysis been performed appropriately and rigorously? 

Reviewer #1: Yes

4. Have the authors made all data underlying the findings in their manuscript fully available?

Reviewer #1: Yes

5. Is the manuscript presented in an intelligible fashion and written in standard English?

Reviewer #1: Yes

6. Review Comments to the Author

Reviewer #1: Dear author(s),

Many thanks for your paying attention and your efforts to revise your manuscript. I also write some new comments. It feels to me it is totally a nice paper. I recommend you implement them. Thank you for your patience:

1. Your abstract is now long, please shorten but do not omit the main part as I said before. In your results part in your abstract, it is dispensable to state all results, just mention to them but with those cautions that I mentioned previously.

2. In your data analysis part, please explicate why p-value<0.2 was chosen or please present an evidence about that. Please cite to a reference.

3. Please state the presumptions of ANOVA models and check them.

4. Please give a reason for making the covariates (income and education level) categorical. It helps you to enhance the validity of your manuscript.

5. please make your tables shipshape in terms of subheadings. (table3 and 4)

6. It would be very very nice if you interpreted the results clearly and please do not confine your paper to just reporting. It will be very nice if you report the measure of effect of each predictor.

7. please write a subtitle for your figure. it is vague.

It is better than the last version but it still needs to revise more.

Regards,

Maryam Ganji

7. PLOS authors have the option to publish the peer review history of their article (what does this mean?). If published, this will include your full peer review and any attached files.

Reviewer #1: No

---

## [Author Response · Author response to Decision Letter 1]

5 Jan 2021

Dear Maryam Ganji,

Thank you very much for your comprehensive comments and constructive suggestions. We read and consider each comment very carefully, and thoroughly revise the manuscript according to your comments and suggestions. We have also edited the language. We hope that the manuscript reads more convincingly after the revision.

Reviewer #1: Dear author(s),

1. Your abstract is now long, please shorten but do not omit the main part as I said before. In your results part in your abstract, it is dispensable to state all results, just mention to them but with those cautions that I mentioned previously.

Response: 

Dear, thank you for the comments, we have revised the abstract to shorten it without losing the main part. Please, see the revised manuscript.

2. In your data analysis part, please explicate why p-value <0.2 was chosen or please present an evidence about that. Please cite to a reference.

Response: 

Thank you very much for the comment. There is a debate about the use of p-value for variable selection. However, often arbitrary choice has to be made about the selection parameter, that is, the significance level to decide whether an effect should be retained in a model. One of the ways to do that is to screen the variables by using p-value such as P value of 0.2 in the bi-variable analysis. 

Evidence: Heinze G, Dunkler D. Five myths about variable selection. Transplant International. 2017;30(1):6-10. This reference has been cited in the data analysis part. Please, see reference number 42.

The main reason why we use the p-value to select variables is to have a parsimonious model, which is a model with the smallest number of variables that have a high effect on the outcome variable and this has its contribution to finding better estimates. The probability of the variables that have a p-value >0.2 from the bi-variable analysis to be significant on the final multivariable model is almost zero. Therefore, we believe adding those variables to the model may affect the parsimony and may also cause overfitting of the model. Moreover,it is usually advised that not just one criterion should be used but if possible an array of criteria. Common model selection criteria are R2, AIC, BIC, and p-level. Therefore, we have used p-value and AIC criteria to get the best final model.

Finally, as we mentioned in the previous response, usage of P value <0.2, during bivariable analysis for variable selection is common practice in our setup. 

3. Please state the presumptions of ANOVA models and check them.

Response: 

Thank you! There are six "assumptions" that underpin the one-way ANOVA. Since the first three assumptions (dependent variable should be measured at the continuous level, the independent variable should consist of two or more categorical, independent (unrelated) groups, and independence of observations) cannot be tested for using software, we have addressed them during study design and choice of variables. But, we can check the rest three assumptions: no significant outliers. dependent variable should be approximately normally distributed, and homogeneity of variances. We have mentioned the three assumptions in the method part of data analysis, and detailed results of the assumptions were given in the result section. Please, see line numbers 132-133 and 181-187.

4. Please give a reason for making the covariates (income and education level) categorical. It helps you to enhance the validity of your manuscript.

Response: 

Dear, thank you for your comments. As we mentioned in the previous response, when the literature review was done, those variables, in most literature, were found as categorical, and for direct comparison purposes, we developed the questionnaire as a closed question (categorical). Unfortunately, we are unable to change those variables into continuous like social support and depression.

5. Please make your tables shipshape interms of subheadings. (table3and4)

Response: 

Thanks for pointing out our shortcomings! We have revised the subheadings of Tables 1, 3 & 4.

6. It would be very very nice if you interpreted the results clearly and please do not confine your paper to just reporting. It will be very nice if you report the measure of effect of each predictor.

Response: 

Dear, thank you for your constructive comments. We have interpreted each coefficient, which was significantly associated with each domain of quality of life. Please, see line numbers 199-207 and 215-220.

7. Please write a subtitle for your figure. It is vague. It is better than the last version but it still needs to revise more.

Response: 

Thank you for your suggestions! we have revised the figure such as subtitle. Please see the revised figure.

---

## [Decision Letter · Decision Letter 2]

1 Mar 2021

Health-related quality of life and associated factors among patients with stroke at tertiary level hospitals in Ethiopia

PONE-D-20-19104R2

Dear Dr. Aschalew,

We’re pleased to inform you that your manuscript has been judged scientifically suitable for publication and will be formally accepted for publication once it meets all outstanding technical requirements.

Kind regards,

Anandh Babu Pon Velayutham

Academic Editor

PLOS ONE

Additional Editor Comments (optional):

Reviewers' comments:

Reviewer's Responses to Questions

**Comments to the Author**

1. If the authors have adequately addressed your comments raised in a previous round of review and you feel that this manuscript is now acceptable for publication, you may indicate that here to bypass the “Comments to the Author” section, enter your conflict of interest statement in the “Confidential to Editor” section, and submit your "Accept" recommendation.

Reviewer #1: All comments have been addressed

Reviewer #2: All comments have been addressed

2. Is the manuscript technically sound, and do the data support the conclusions?

Reviewer #1: Yes

Reviewer #2: Yes

3. Has the statistical analysis been performed appropriately and rigorously? 

Reviewer #1: Yes

Reviewer #2: Yes

4. Have the authors made all data underlying the findings in their manuscript fully available?

Reviewer #1: Yes

Reviewer #2: Yes

5. Is the manuscript presented in an intelligible fashion and written in standard English?

Reviewer #1: Yes

Reviewer #2: Yes

6. Review Comments to the Author

Reviewer #1: Dear author(s),

Congratulation! your paper is accepted and reached the merit of publication criteria in PLOS ONE journal. Thank you for cooperation, patience and edition. Bravo, you nailed it.

Best,

Maryam Ganji

Reviewer #2: In the second revised version all reviewer's comments have been addressed satisfactorily and the quality of this manuscript has been improved.

7. PLOS authors have the option to publish the peer review history of their article (what does this mean?). If published, this will include your full peer review and any attached files.

Reviewer #1: **Yes: **Maryam Ganji

Reviewer #2: No

---

## [Editor Report · Acceptance letter]

8 Mar 2021

PONE-D-20-19104R2 

Health-related quality of life and associated factors among patients with stroke at tertiary level hospitals in Ethiopia 

Dear Dr. Yalew Aschalew:

I'm pleased to inform you that your manuscript has been deemed suitable for publication in PLOS ONE. Congratulations! Your manuscript is now with our production department. 

Kind regards, 

on behalf of

Dr. Anandh Babu Pon Velayutham 

Academic Editor

PLOS ONE